# The Role of Iron in Calciphylaxis—A Current Review

**DOI:** 10.3390/jcm11195779

**Published:** 2022-09-29

**Authors:** Olivia Wickens, Sharmilee Rengarajan, Rajkumar Chinnadurai, Ian Ford, Iain C. Macdougall, Philip A. Kalra, Smeeta Sinha

**Affiliations:** 1Salford Royal Hospital, Northern Care Alliance NHS Foundation Trust, Salford M6 8HD, UK; 2Faculty of Biology, Medicine and Health, University of Manchester, Manchester M13 9PL, UK; 3Robertson Centre for Biostatistics, University of Glasgow, Glasgow G12 8QQ, UK; 4Department of Renal Medicine, King’s College Hospital, London SE5 9RS, UK

**Keywords:** iron, calcific uraemic arteriolopathy, calciphylaxis, haemodialysis, end-stage renal disease

## Abstract

Calcific uraemic arteriolopathy (CUA), also known as calciphylaxis, is a rare and often fatal condition, frequently diagnosed in end-stage renal disease (ESRD) patients. Although exact pathogenesis remains unclear, iron supplementation is suggested as a potential risk factor. Iron and erythropoietin are the main stay of treatment for anaemia in ESRD patients. Few observational studies support the role of iron in the pathogenesis of calciphylaxis although data from the pivotal trial was not strongly supportive of this argument, i.e., no difference in incidence of calciphylaxis between the low-dose and high-dose iron treatment arms. Elevated levels of vascular cell adhesion molecules in association with iron excess were postulated to the pathogenesis of CUA by causing inflammation and calcification within the microvasculature. In-addition, oxidative stress generated because of iron deposition in cases of systemic inflammation, such as those seen in ESRD, may play a role in vascular calcification. Despite these arguments, a direct correlation between cumulative iron exposure with CUA incidence is not clearly demonstrated in the literature. Consequently, we do not have evidence to recommend iron reduction or cessation in ESRD patients that develop CUA.

## 1. Introduction

Calcific uraemic arteriolopathy (CUA), often referred to as calciphylaxis, was initially described by Bryant and White in 1898 [1]. It is a very serious and frequently fatal condition of cutaneous microvascular calcification and thought to be an independent risk factor linked to all-cause mortality [2,3]. Although rare, it is most commonly documented in patients with chronic kidney disease (CKD), in particular those with end-stage renal disease (ESRD), affecting up to 1–4% of ESRD patients, associated with high morbidity and mortality [4,5,6] Once diagnosed, prognosis is often poor, even with treatment, with an associated mortality of around 45–80% at one year [2,7]. CKD patients are at increased risk for developing vascular calcification due to their associated uraemia which predisposes to inflammation and endothelial dysfunction, although it has rarely been reported in patients without evidence of renal dysfunction, termed non-uraemic CUA [2,8,9].

CUA can have a wide spectrum of different clinical presentations including painful cutaneous plaques or purpura, erythematous lesions, livedo racemosa, subcutaneous nodules, excruciatingly painful ischaemic, necrotic skin ulceration and subsequent gangrene [2,7,8,10,11]. A frequent complication, and major cause of mortality, in patients with CUA is that of secondary skin infection, which can result in septicaemia and death [11].

Although the exact pathogenesis of CUA is undetermined, it is thought to result from extensive microvascular calcification and thrombosis within the media of arterioles, occlusion of which can result in necrosis of the overlying skin and soft tissues [5,6,8]. The precise pathophysiology of microvascular calcification is unclear; it is suggested that there is likely an underlying imbalance between promoters and inhibitors of vascular calcification in those who develop this condition [8]. Numerous risk factors are suggested in its development; namely ESRD, female gender, Caucasian ethnicity, diabetes mellitus, obesity or significant rapid weight loss, local trauma, abnormalities in coagulation such as protein C or S deficiency, autoimmune disorders, recurrent hypotension, hyperparathyroidism, hypercalcaemia, hyperphosphataemia, elevated alkaline phosphatase, hypoalbuminaemia and exposure to ultraviolet light [2,5,7,8,9,12,13]. Additionally, there are several medications which have been associated with increased risk, such as, vitamin K antagonists, calcium, calcium-containing phosphate binders, phosphate, erythropoiesis-stimulating agents (ESAs), iron, vitamin D supplements and immunosuppressants, such as methotrexate and corticosteroids [5,6,7,12]. In a previous study it was found that vitamin K antagonist use, high C-reactive protein, low albumin and low haemoglobin in association with CUA were markers of poor prognosis similar to observations from a French cohort [3,14].

## 2. Role of Iron in Calciphylaxis

Although the aetiology of CUA is likely to be multifactorial, evidence in literature have suggested a possible role for iron in its pathogenesis. Iron is an essential component of haemoglobin and several metalloproteins and plays a crucial role in a variety of cellular functions fundamental for life [15]. Anaemia is a frequent complication of ESRD and patients are often iron deplete. Iron supplementation along with ESAs form the mainstay of treatment in such patients, with the majority of patients likely to have been exposed to iron at some point. Although essential for life, excess iron accumulation can result in cell death, tissue damage and organ dysfunction through the production of reactive oxygen species (ROS) and oxidative stress; it can also lead to an impaired response to infection and inflammation [15,16,17].

In 1962 Hans Selye [18] was the first person to name CUA as a condition of induced systemic hypersensitivity, which he termed calciphylaxis. He was also the first to suggest a causal link for iron in its development based on his ‘two-hit hypothesis’ from research in rodents that revealed calcium deposition would result if initially exposed to “sensitisers” such as vitamin D or parathyroid hormone and then were subsequently exposed to chemical or mechanical “challengers” such as iron, egg whites or egg yolk [1,8,19]. This was reinforced by an early case report in 1967, which documented widespread calcification which developed over the lateral and anterior aspect of both thighs of a patient following intramuscular injection of iron dextran complex administered into the same area [20]. A further two patients with vascular calcification and tissue gangrene were described by Rubinger et al. [21] in 1986, both of which were receiving haemodialysis (HD) and had severe secondary hyperparathyroidism. Iron overload was evident in both patients, with histology of the digital arteries of one patient revealing the presence of iron which stained strongly positive amongst the periphery of areas of calcification. However, it is worth mentioning that both patients had received multiple blood transfusions rather than iron supplementation, with one patient requiring more than 100 blood transfusions with evidence of massive iron deposition within a liver biopsy and bone marrow sample.

Another study by Amuluru et al. [6] revealed significantly higher quantities of iron in all 12 tissues samples from patients with a histological diagnosis of CUA compared to skin biopsies from patients without CUA used as controls. The five control samples were taken from patients with CKD who had been exposed to gadolinium contrast. Out of these 12 patients with CUA, four were reported to have a history of iron supplementation (either parenteral or oral) prior to the diagnosis of CUA, one patient had not received any iron supplementation, and the remainder were unknown. Laboratory data was only available for five CUA patients, with no data available for control patients. In this study the control tissue biopsies were taken from patients with CKD stage 3 and not those with ESRD, which might have meant they were less likely to have been exposed to iron and a lesser uraemic state than that in patients with ESRD.

Additionally, Farah et al. [8] visualised iron in all 12 CUA biopsies taken from patients with ESRD, of which nine patients were reported to have received exposure to some form of iron supplementation, although the quantity and duration of iron supplementation was not apparent. Of particular interest, was that iron has been reported not only within the affected arterioles of CUA tissues specimens, but also in the area outside of vessels and, in addition, identified in extravascular tissue alone, with no obvious explanation [8].

Although Amuluru et al. [6] and Farah et al. [8] revealed iron deposition within tissue samples from CUA lesions and Rubinger et al. [21] in two patients with vascular calcification, without a complete history of iron exposure, blood transfusions and serum iron studies it is difficult to infer whether iron deposition is correlated with increased iron exposure or iron overload.

Two studies by Panchal et al. [5] and Zacharias at al. [13] which investigated serum iron and iron exposure in CUA did not reveal a significant association between iron and CUA. Panchal et al. [5] revealed that out of 30 patients who developed CUA, of which 28 had a medication history available at diagnosis, five (18%) patients were taking iron supplements. All patients with CUA had low or normal serum iron levels, with an elevated ferritin level in 75% patients. However, as ferritin is not only a marker of iron stores, but also an acute phase reactant, it is usually elevated in the presence of inflammation, which can occur in CUA and ESRD and as such, is not a reliable marker of iron stores. Zacharias at al. [13] reported eight continuous ambulatory peritoneal dialysis (CAPD) patients who had a diagnosis of CUA, in which laboratory data and cumulative dose of ferrous sulphate were collected for the year prior to development of CUA in the index cases. Each case of CUA was matched with up to five control cases and comparison between the 8 CUA cases and 37 controls revealed equal iron levels and iron saturation between the groups. Multivariate analysis revealed iron usage up to six months prior to CUA development was revealed to be a protective factor. Administration of intravenous iron dextran was recorded in two patients; however, this was not within six months of them developing CUA and there was no temporal correlation between intravenous iron usage and CUA onset. 

Liver hepatocytes are the main storage site for body iron and interestingly, Rostoker et al. [22] found that on analysis of liver iron concentration (LIC) in 358 dialysis patients, measured by quantitative magnetic resonance imaging (qMRI), 243 dialysis patients had mild, moderate or severe radiologic iron overload, of which only one patient with mild LIC overload developed CUA. There were no reported cases of CUA in any of the patients with normal, moderate or severe levels of LIC. Thus, there is no strong positive correlation with LIC and the development of CUA, supporting the evidence that the development of CUA is based upon a multitude of factors, especially with the incidence of CUA remaining low despite so many patients being exposed to iron therapy with associated iron overload. 

Although there is no cure for CUA, treatments include intensive dialysis, hyperbaric oxygen, parathyroidectomy, aggressive wound care and cessation of possible drug causes [7,11]. With iron supplementation being a crucial component of anaemia management, cessation of iron treatment has important implications for patients due to the adverse effects associated with anaemia. It was demonstrated in the pivotal trial that high-dose iron was superior to that of low-dose iron, with a reduced incidence of death or major cardiovascular events [23]. Consequently, stopping iron treatment could result in worsened patient outcomes and result in more harm than good. An alternative to this as suggested by Rostoker et al. [22] may not require iron cessation, but instead close monitoring of liver iron storage via qMRI, which is the most important organ for iron storage [16].

## 3. Mechanistic Insights on the Role of Iron in the Pathogenesis of Calciphylaxis

In addition to the findings of iron deposition within CUA specimens, there have been several proposed theories as to why iron might be associated with CUA pathogenesis, such as its known toxic effect in overload, including iron-induced oxidative stress, and its role in conditions such as atherosclerosis and vascular calcification. Research into atherosclerosis has revealed elevated levels of vascular cell adhesion molecules (VCAMs) and an increase in intima-media thickness with iron excess or supplementation [24]. Iron accumulation is also found within atherosclerotic plaques, particularly those which are symptomatic [24]. Increased quantities of soluble adhesion molecules were found in CKD patients who received iron sucrose, with cumulative iron dose correlating with increased intima-media thickness [24]. It is therefore possible that VCAMs, which have been associated with vascular calcification, may contribute to the pathogenesis of CUA by causing inflammation and calcification within the microvasculature [25]. Iron deposition might also occur in cases of systemic inflammation, such as those seen in ESRD, which may generate catalytic iron at the tissue level [6].

It is known that oxidative stress plays a key role in the development of atherosclerosis, but it is only more recently identified that oxidative stress plays a role in the development of vascular calcification [6,26]. When ESRD patients are administered iron compounds they dissociate in the reticuloendothelial system (RES) where a number of different iron transfers take place. As a consequence of these, labile iron can be released into the circulation which could result in the formation of ROS, iron-induced oxidative stress and endothelial injury [6,26]. Oxidative stress can cause a phenotypic switch from a contractile to osteochondrogenic vascular smooth muscle cells (VSMC), a transdifferentiation which is thought to be associated with the pathogenesis of CUA and vascular calcification [6,26]. This can contribute to the already increased uraemic state of oxidative stress and inflammation observed in patients with ESRD, further heightening vascular calcification. Additionally, ESRD patients receiving HD are at risk of abnormal iron metabolism and increased intravenous iron exposure. Intravenous iron can bypass the intestinal absorption system regulated by hepcidin, which consequently may result in free iron deposition to occur in sensitised areas of microvasculature, potentially mediating vascular calcification [8,27]. The management of calciphylaxis is a multimodal approach with promising results reported using sodium thiosulphate (STS). It is important to note that STS is postulated to act as an antioxidant and vasodilator, thereby preventing the calcification of blood vessels [28,29,30].

Iron has also been suggested to be involved in initiation of the nucleation process in crystal deposition that may contribute to precipitation of calcium and phosphorus within lesions [6]. However, it should be mentioned that along with its effects in promoting vascular calcification, there have also been inhibitory effects of iron on VSMC calcification reported [26] (Figure 1).

Macrophages play a key role in iron regulation and in preventing toxicity. In conditions such as chronic venous disease (CVD), there is erythrocyte extravasation secondary to venous hypertension, leading to haemosiderin deposition and enabling the activation of macrophages which become loaded with the iron via erythrophagocytosis [31]. Studies into CVD revealed a particular subtype of M1 pro-inflammatory macrophages present in areas of increased iron deposition, resulting in high pro-inflammatory cytokine production and reduced wound healing [32,33]. In a murine study, mice repeatedly exposed to iron dextran, which accumulated within the dermis, were then wounded revealing delayed wound healing and greater accumulation and persistence of macrophages compared to controls [32]. After wounding there was accumulation of macrophages with iron and evidence of an unrestrained pro-inflammatory M1 population of macrophages that were highly expressed in the mice repeatedly exposed to iron dextran [32]. Although erythrocyte extravasation is rare in CUA, chronic iron administration and abnormal iron kinetics may cause increased macrophage iron sequestration, which along with iron deposition could, in theory, cause activation of this pro-inflammatory subtype of M1 macrophages resulting in poor wound healing, high infection rates and worsened prognosis of CUA lesions [6,8]. As iron is associated with an impaired response to infection, the addition of M1 pro-inflammatory macrophage activation as a result of iron overload, may explain why CUA is so challenging to treat. M1 macrophages are also associated with the release of pro-inflammatory cytokines which are associated with greater severity and progression of coronary vascular calcification in patients receiving HD [34]. 

In addition to iron overload, iron deficiency and anaemia of inflammation seen in ESRD may also contribute to poor wound healing and deposition of iron within macrophages as a result of raised hepcidin levels and subsequent interaction between hepcidin and ferroportin [31].

Although iron is implicated in various vascular pathologies, there exist beneficial cellular responses to iron for affecting vascular mineralization [35]. Iron exposure of cells of vasculature and heart valves including vascular smooth muscle cells, valvular interstitial cell and mesenchymal stem cells, inhibits their osteochondrogenic transition and calcification in calcifying condition [35,36,37]. The inhibition is mediated by H-ferritin/ferroxidase activity up-regulated by iron in these cells. The regulation of osteoblast activity by H-ferritin occurs via the control of cellular phosphate uptake, activities of Runt-related transcription factor 2 (RUNX2) and Sox9 (SRY (sex-determining region Y)-box 9), sequestration of lysosomal phosphate, and pyrophosphate generation [35,37]. Such cellular responses to iron might contribute to the dichotomy attributed to H-ferritin in vascular calcification. 

Even though several studies have revealed iron deposition within CUA lesions, it would be very useful to compare these to patients never having been exposed to iron or blood transfusions. However, in the ESRD population this would be extremely challenging. Further studies are required to obtain the quantity, duration and method of administration of iron exposure in patients with CUA, along with iron quantification in tissue biopsies which would enable analysis of potential correlation between iron exposure and iron deposition.

## 4. Pivotal Randomised Controlled Trial Data and Its Implications

With several studies suggesting a potential role of iron as a risk factor for the development of CUA, the data of patients who developed CUA on follow up in the pivotal trial [23], were identified and investigated. The pivotal study, was a multicentre, open-label trial with blinded end-point evaluation, in which maintenance HD patients were randomly assigned to receiving either high-dose intravenous iron sucrose administered proactively (400 mg per month) to target serum ferritin up to 700 μg/L, or low-dose iron sucrose administered intravenously in a reactive fashion (0 to 400 mg monthly) to maintain serum ferritin at 200 μg/L across 50 sites in the United Kingdom. In pivotal, patients with ESRD were recruited if they had commenced HD no longer than 12 months prior to the initial screening visit and had a ferritin concentration of less than 400 μg per liter as well as a transferrin saturation of less than 30%.

A total of 2141 patients were recruited into the pivotal trial, and they were followed up over a median period of 2.1 years. This trial commenced in November 2013 and patients were recruited across 50 sites within the United Kingdom. There were 1093 patients who were randomised to the high-dose iron group and 1048 patients to the low-dose iron group. The median monthly dose of iron sucrose in the low-dose iron group was 145 mg (interquartile range (IQR) 100–190) and in the high-dose group it was 264 mg (IQR 200–336).

Of the six patients who developed CUA, 50% (*n* = 3) were reported in the low-dose intravenous iron group and 50% (*n* = 3) were reported in the high-dose intravenous iron group. The number of patients who developed CUA was 0.29% in the low-dose group and 0.27% in the high-dose group. Of the total number of patients with a diagnosis of CUA there was an equal gender distribution. The mean age at the time of diagnosis was 64 years (range 50–82 years).

There are varying reported incidence rates of CUA across the world ranging from as high as 45 per 1000 patient years amongst HD and peritoneal dialysis (PD) patients in Canada [38] to possibly as low as <0.1 per 1000 patient years in Japan [39]. The incidence rate of 2.8 cases in 1000 patient years that we observed in the pivotal trial is lower than the mean reported CUA incidence which is about 3.49 per 1000 patient years [22,40]. Additionally, the observed incidence of CUA was lower than that reported in the Evaluation of Cinacalcet Hydrochloride Therapy to Lower Cardio Vascular Events (EVOLVE) trial which reported a CUA incidence rate of 3 per 1000 patient years [12].

The pivotal data also revealed equal incidence of CUA diagnosed in both the high and low-dose iron groups. Although, we unfortunately did not have a cohort of patients that were not receiving intravenous iron for direct comparison. 

It should also be mentioned that the population of patients that were recruited into the pivotal study were those with a shorter HD duration (less than 12 months prior to first screening assessment) and lower ferritin levels (ferritin less than 400 μg per L). Thus, the lower incidence of CUA may have been explained by the lower levels of inflammation and shorter duration of HD amongst this population.

## 5. Conclusions

In summary, the pathogenesis of CUA remains speculative. With several reports of iron being demonstrated within CUA lesions and the toxic effects of surplus iron it is understood why iron could play a potential role in CUA pathogenesis, and although there is no evidence of direct causation, iron may contribute to a favourable milieu. However, there are no studies to date which have provided a direct correlation between cumulative iron exposure with CUA incidence. As such, from our review of the current literature surrounding iron and CUA, we do not have the evidence to recommend a reduction or cessation of iron therapy in patients requiring treatment who develop CUA. Iron deficiency can impact negatively upon patients, particularly in those with cardiovascular disease, which is often the leading cause of death in patients with ESRD, consequently causing more harm than good. It is also not understood how iron accumulation and overall iron burden within CUA lesions changes with time, which would be useful in determining whether this occurs as a primary or secondary event of microvascular calcification. Further research is required to investigate the role of iron and CUA.

## Figures and Tables

**Figure 1 jcm-11-05779-f001:**
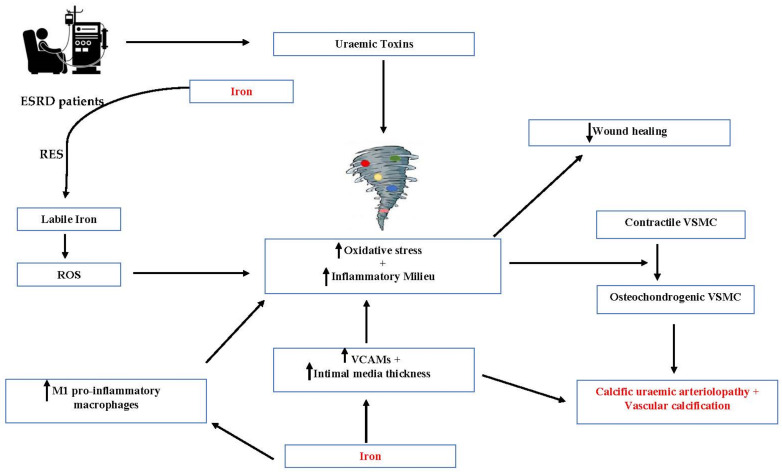
Potential pathophysiological mechanisms of Iron in Calciphylaxis. ESRD—end stage renal disease, RES—reticulo-endothelial system, ROS—reactive oxygen species, VCAMs—vascular cell adhesion molecules, VSMC—vascular smooth muscle cells.

## Data Availability

Not applicable.

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
