# Peer review of "The Role of Iron in Calciphylaxis—A Current Review"

_jcm, 2022, doi:10.3390/jcm11195779_

Round 1

Reviewer 1 Report

The review is well-written and a comprehensive review. However, some corrections/additions are suugested:

1. Livedo reticularis should be replaced by livedo racemosa (livedo reticularis is a benign phenomenon

2. Overall, some of the references are quite old. The recent reference by Rick J et al in J Am Acad Dermatol May 2022 should be read by the authors and cited in an appropriate manner.

3. The role of sodium thiosulphate in treatment is totally missing, and should be added and discussed. Also there could be some discussion of the role of sodium thiosulphate as an antioxidant in the section where the role of iron is discussed as a generator of ROS.

Author Response

The review is well-written and a comprehensive review. However, some corrections/additions are suggested:

Q1. Livedo reticularis should be replaced by livedo racemosa (livedo reticularis is a benign phenomenon.

Answer: Thank you. We have now made this change as recommended (line-39).

Q2. Overall, some of the references are quite old. The recent reference by Rick J et al in J Am Acad Dermatol May 2022 should be read by the authors and cited in an appropriate manner.

Answer: Thank you. We have now incorporated this reference (line 181).

Q3. The role of sodium thiosulphate in treatment is totally missing, and should be added and discussed. Also there could be some discussion of the role of sodium thiosulphate as an antioxidant in the section where the role of iron is discussed as a generator of ROS.

Answer: Thank you very much for this comment. We have now included the role of sodium thiosulphate as an antioxidant at the appropriate section as indicated (lines- 178 to181).

Reviewer 2 Report

The manuscript by Olivia Wickens et al. summarizes the current knowledge on the role of iron in calciphylaxis. Calciphylaxis is still a life threatening pathology of arterioles primarily affected CKD patients at stage V receiving renal replace therapy. The lack of specific treatment put an emphasis on the exploration of possible contributing factors including iron. Since such a patient population regularly receives iron supplementation i.v. the aim of the review has paramount importance. The focus is on both the clinical studies and the molecular mechanism of harm associated with iron within the vasculature possibly connected to calcification. This is a great review and well written. Authors, Sinha’s research group, have significant contribution to the research field.

Comments:

Authors summarizes the possible molecular mechanisms associated with iron excess that may contribute to the development of CUA. They conclude that direct correlation between cumulative iron exposure with CUA incidence has not been demonstrated by clinical studies.

Although iron is implicated in various vascular pathologies there exist beneficial cellular responses to iron for affecting vascular mineralization (PMID: 19423691). Iron exposure of cells of vasculature and heart valves including vascular smooth muscle cells, valvular interstitial cell and mesenchymal stem cells inhibits their osteochondrogenic transition and calcification in calcifying condition (PMID: 19423691, PMID: 30700131, PMID: 27287253). The inhibition is mediated by H-ferritin/ferroxidase activity up-regulated by iron in these cells. The regulation of osteoblast activity by H-ferritin occurs via the control of cellular phosphate uptake, activities of Runt-related transcription factor 2 (RUNX2) and Sox9 [sex-determining region Y]-box 9), sequestration of lysosomal phosphate, and pyrophosphate generation (PMID: 19423691, PMID: 30700131). Such cellular responses to iron might contribute to the dichotomy attributed to H-ferritin in vascular calcification.

These findings may improve the merit of the manuscript.

Author Response

Q1: Comments:

Authors summarizes the possible molecular mechanisms associated with iron excess that may contribute to the development of CUA. They conclude that direct correlation between cumulative iron exposure with CUA incidence has not been demonstrated by clinical studies.

Although iron is implicated in various vascular pathologies there exist beneficial cellular responses to iron for affecting vascular mineralization (PMID: 19423691). Iron exposure of cells of vasculature and heart valves including vascular smooth muscle cells, valvular interstitial cell and mesenchymal stem cells inhibits their osteochondrogenic transition and calcification in calcifying condition (PMID: 19423691, PMID: 30700131, PMID: 27287253). The inhibition is mediated by H-ferritin/ferroxidase activity up-regulated by iron in these cells. The regulation of osteoblast activity by H-ferritin occurs via the control of cellular phosphate uptake, activities of Runt-related transcription factor 2 (RUNX2) and Sox9 [sex-determining region Y]-box 9), sequestration of lysosomal phosphate, and pyrophosphate generation (PMID: 19423691, PMID: 30700131). Such cellular responses to iron might contribute to the dichotomy attributed to H-ferritin in vascular calcification. These findings may improve the merit of the manuscript.

Answer: Thank you very much for your comments. We have included these points to our mechanistic insights sub-heading (lines- 216-226). These points certainly improve the strength of our discussions. Many thanks again.